



# Cirrus formation regimes - Data driven identification and quantification of mineral dust effect

Kai Jeggle[1], David Neubauer[1], Hanin Binder[1], and Ulrike Lohmann[1]

[1]Institute for Atmospheric and Climate Science, ETH Zurich, Zurich, Switzerland

**Correspondence:** Kai Jeggle (kai.jeggle@env.ethz.ch), Ulrike Lohmann (ulrike.lohmann@env.ethz.ch)

**Abstract.** The microphysical and radiative properties of cirrus clouds are strongly dependent on the ice nucleation mechanism and origin of the ice crystals. Due to sparse temporal coverage of satellite data and limited observations of ice nucleating particles (INPs) at cirrus levels it is notoriously hard to determine the origin of the ice and the nucleation mechanism of cirrus clouds in satellite observations. In this work we combine three years of satellite observations of cirrus clouds from the DARDAR-Nice retrieval product with Lagrangian trajectories of reanalysis data of meteorological and aerosol variables calculated 24 h backward in time for each observed cirrus cloud. In a first step, we identify typical cirrus cloud formation regimes by clustering the Lagrangian trajectories and characterize observed microphysical properties for in situ and liquid origin cirrus clouds in midlatitudes and the tropics. On average, in situ cirrus clouds have smaller ice water content (IWC) and lower ice crystal number concentration ($N_{ice}$) and a strong negative temperature dependence of $N_{ice}$, while liquid origin cirrus have a larger IWC and higher $N_{ice}$ and a strong positive temperature dependence of IWC. In a second step, we use MERRA2 reanalysis data to quantify the sensitivity of cirrus cloud microphysical properties to a change in the concentration of dust particles that may act as INPs. By identifying similar cirrus cloud formation pathways, we can condition on ice-origin, region, and meteorological dependencies, and quantify the impact of dust particles for different formation regimes. We find that at cloud top median $N_{ice}$ decreases with increasing dust concentrations for liquid origin cirrus. Specifically, the sensitivities are between 5 % and 11 % per unit increase of dust concentration in logarithmic space in the tropics and between 12 % and 18 % in the mid-latitudes. The decrease in $N_{ice}$ can be explained by increased heterogeneous ice nucleation in the mixed-phase regime, leading to fewer cloud droplets freezing homogeneously once the cloud enters the cirrus temperatures and glaciates. The resulting fewer, but larger ice crystals are more likely to sediment, leading to reduced IWC, as for example observed for liquid origin cirrus in the mid-latitudes. In contrast, for in situ cirrus in the tropics, we find an increase of $N_{ice}$ median values of 21 % per unit increase of dust aerosol in logarithmic space. We assume that this is caused by heterogeneous nucleation of ice initiated by dust INPs in INP limited conditions with supersaturations between the heterogeneous and homogeneous freezing thresholds. Such conditions frequently occur at high altitudes, especially in tropical regions at temperatures below 200 K.

Our results provide an observational line of evidence that the climate intervention method of seeding cirrus clouds with potent INPs may result in an undesired positive cloud radiative effect (CRE), i.e. a warming effect. Instead of producing fewer but larger ice crystals, which would lead to the desired negative CRE, we show that additional INPs can lead to an increase in $N_{ice}$, an effect called *overseeding*.



## 1 Introduction

Cirrus clouds have a regional-dependent occurrence between 10 % and 50 %, with an increasing occurrence towards the equator (Heymsfield et al., 2017). Despite the high cirrus cloud cover and the resulting climatological relevance, cirrus are a source of large uncertainties in climate projections (Forster et al., 2021). Cirrus clouds occur primarily in the upper troposphere at temperatures below -38°C and consist purely of ice crystals. Clouds modulate the Earth's radiative budget via the reflection of shortwave radiation (cloud albedo effect) and the absorption and emission of infrared radiation into space (cloud greenhouse effect). Global climate model (GCM) and observational studies show that the greenhouse effect dominates in cirrus clouds, resulting in a positive CRE, meaning a net warming of the atmosphere (Gasparini and Lohmann, 2016; Hong et al., 2016). However, depending on the cloud microphysical properties (CMP) of cirrus, the magnitude of the CRE varies significantly and can even change to a negative CRE for lower-level optically thick cirrus (DeMott et al., 2010; Hong et al., 2016; Krämer et al., 2016; Heymsfield et al., 2017). Cirrus CMP, namely the IWC and $N_{ice}$, are in turn controlled by the competition between homogeneous and heterogeneous ice nucleation (e.g. Kärcher et al., 2006), ice origin (Krämer et al., 2016), geographical location (Heymsfield et al., 2017), environmental variables, mainly temperature and updraft velocities, as well as the aerosol environment (Gryspeerdt et al., 2018). The heterogeneity of cirrus CMP and subsequent CRE require a better understanding of the formation pathways and their influence on cirrus CMP to reduce the uncertainties in CRE estimates and cirrus cloud representations in GCMs used for climate change projections.

In recent years there have been significant advances in the understanding of cirrus cloud formation and the resulting CMP using in situ aircraft data, global satellite data, as well as model simulations. Krämer et al. (2016) and Luebke et al. (2016) introduced a cirrus classification based on the origin of ice crystals in cirrus clouds. Cirrus that form directly from the gas phase at temperatures $T < -38°C$ are denoted in situ cirrus. Their counter part are liquid origin cirrus clouds. These clouds evolve from mixed phase clouds where cloud droplets freeze heterogeneously via ice nucleating particles (INP) (Kanji et al., 2017) at temperatures $T > -38°C$. Once the cloud is lifted to temperatures $T < -38°C$ it glaciates spontaneously by homogeneous freezing of the remaining cloud droplets. Krämer et al. (2016) combined in situ aircraft observations with model simulations to study the CMP associated with the two ice-origin regimes. The authors find that in situ cirrus form at higher altitudes, are optically thinner and consist of fewer ice crystals. Liquid origin cirrus on the other hand are thicker clouds found at lower altitudes consisting of many ice crystals. Their study suggests that liquid origin clouds have a cooling and in situ a warming effect on the atmosphere.

Multiple other studies have adopted the classification of cirrus with respect to ice origin. For instance, Wernli et al. (2016) analyzed 12 years of ERA-Interim trajectories in the North Atlantic to identify the relative occurrence of liquid origin and in situ cirrus; Gasparini et al. (2018) compared satellite observations from the CALIPSO satellite (Hunt et al., 2009) with simulation data from GCMs to study CMP of in situ and liquid origin cirrus. While data from active satellite instruments like CALIPSO's lidar provide an unprecedented vertically resolved global view on real cirrus clouds, the observations suffer from the long revisiting time of the satellite of 16 days which only allows for a temporal snapshot perspective on cirrus clouds. To still approximate the ice origin of satellite observed cirrus. Gasparini et al. (2018) used a heuristic approach to determine ice





origin, in which cirrus are considered liquid origin if they extend to temperatures warmer than -35°C. However, due to the temporal snapshot perspective, anvil cirrus and frontal cirrus are falsely classified as in situ cirrus in this approach. The above mentioned studies all agree that liquid origin cirrus clouds occur at lower altitudes and contain more IWC and higher $N_{ice}$ than in situ cirrus clouds, which occur at higher altitudes and are typically thinner with less IWC and lower $N_{ice}$.

Apart from studying cirrus ice-origin, the relationships between meteorological and aerosol cirrus drivers and CMP have also been studied extensively using observational data. Gryspeerdt et al. (2018) found that next to ice-origin, mainly temperature, updraft by controlling supersaturation, and the aerosol environment determine the CMP. Aerosols can influence CMP by acting as INPs for heterogeneous freezing. In addition to temperature and particle morphology, the ability of INPs to initiate heterogeneous freezing in cirrus clouds is strongly dependent on ambient supersaturations. Recent analysis of 65 airborne relative

humidity measurements at cirrus levels have shown that both in midlatitudes and the tropics, regions with supersaturations between the homogeneous and heterogeneous freezing thresholds exist both in cloudy and cloud-free regions (Krämer et al., 2020; Dekoutsidis et al., 2023). We refer to these conditions as INP limited, as the availability of (additional) INPs could induce heterogeneous freezing and hence increase $N_{ice}$ in existing clouds and also form new clouds. The frequency of such conditions is more prevalent in tropical regions below 200 K. Mineral dust is considered the most important INP in the atmosphere (Kanji

et al., 2017). Kuebbeler et al. (2014) found in a GCM-based study that an increase in dust INP leads to a global reduction of ice mass and an increase of crystal size, caused by a partial switch from homogeneous to heterogeneous nucleation. The exact effect of INPs on cirrus in observed cirrus is still uncertain and difficult to study in observational data as INPs generally only represent a small subset of the overall aerosol population (Lacher et al., 2018; Forster et al., 2021) and the INP effect may be masked by other dependencies such as regional influences, meteorological conditions, and ice-origin.

An additional motivation to study the impact of INPs on cirrus cloud properties is the assessment of the climate intervention method of cirrus cloud thinning (CCT) which aims at altering the radiative properties of cirrus clouds to counteract global warming. CCT was first proposed by Mitchell and Finnegan (2009), building on results by Lohmann et al. (2008) who found in a modeling study that a change of the dominant ice nucleation mechanism from homogeneous to heterogeneous nucleation leads to a large reduction of the cirrus CRE by -2.0 $Wm^{-2}$. The idea behind CCT is to induce a shift of the dominant nucleation

mechanism by injecting INPs into cirrus regimes with the aim of *producing* fewer, but larger ice crystals, which reflect less shortwave radiation, but more importantly, trap less longwave radiation. Storelvmo et al. (2013) simulated CCT in a global climate model and found two opposing effects. A substantial cooling effect was achieved when injecting the *optimal* amount of INPs into cirrus regimes, namely 18 $L^{-1}$. However, at higher concentrations of injected INPs more and smaller ice crystals were observed, which resulted in a substantial warming effect. This undesirable response to CCT is termed *overseeding*. Mul-

tiple other studies with different global climate models and seeding strategies report a negligible cooling potential of CCT, but confirm an *overseeding* effect at high seeding concentrations (Penner et al., 2015; Gasparini and Lohmann, 2016; Gasparini et al., 2017; Tully et al., 2022).

   In this study we combine instantaneous satellite observations with Lagrangian backward trajectories of meteorological and aerosol reanalysis data on a large regional and temporal scale. With this approach, we extend the temporal snapshots obtained

by satellite observations with an evolutionary perspective on cloud formation and development. Our goal is to classify observed



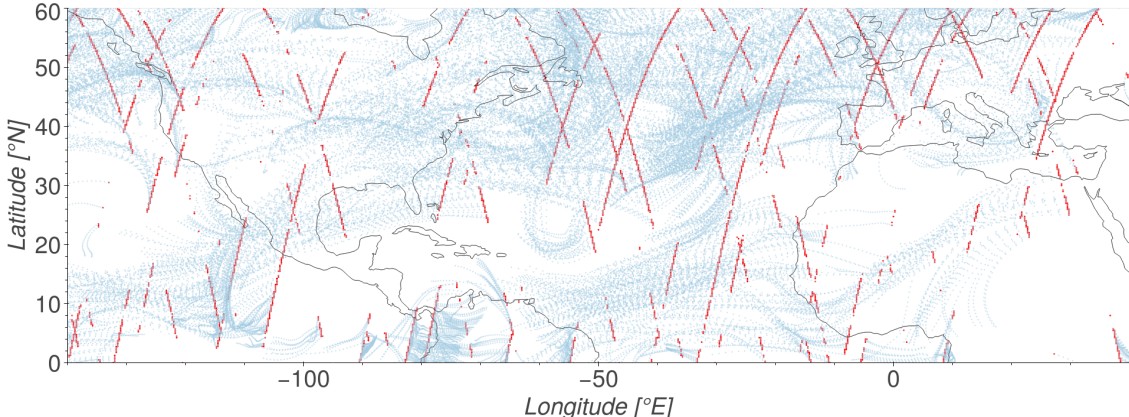

**Figure 1.** Visualization of 10.000 randomly sampled cirrus cloud observations in the study domain (red dots) with their corresponding 24 h Lagrangian backward trajectories (blue dots). For visualization purposes only the trajectories of the cloud top layers are shown.

cirrus clouds by means of their trajectories and to investigate the influence of typical formation pathways on observed cirrus CMP. We follow a data-driven approach by using k-means clustering of Lagrangian trajectories with dynamic time warping (DTW) as distance metric to identify typical cirrus formation regimes. Our second objective is to quantify the effect of mineral dust particle concentrations on cirrus CMP. By identifying cirrus clouds that have formed in similar meteorological environments in midlatitudinal and tropical regions in the first step, we are able to disentangle the influence of ice-origin, meteorological conditions, and regional occurrence from the dust effect.

## 2 Methods

### 2.1 Data

In this work, three years of satellite observations of cirrus clouds are extended with Lagrangian backward trajectories of meteorological variables and aerosol properties. DARDAR (Delanoë and Hogan, 2008) and its extension DARDAR-$N_{ice}$ (Sourdeval et al., 2018) are widely used satellite products that provide vertically resolved retrievals for IWC and $N_{ice}$. DARDAR is a synergistic product combining CloudSat's radar with CALIPSO's lidar and is hence able to penetrate through deep convective clouds, while still being sensitive to optically thin cirrus clouds. Due to the long 16-day revisiting times, DARDAR only provides temporal snapshots along its narrow satellite overpass. The locations of observed cirrus for 10.000 randomly sampled cirrus clouds, representing approximately 1% of the dataset used in this study, are visualized as red dots in Fig. 1. The satellite overpasses are clearly detectable in the figure.

To add a notion of temporal development to observed cirrus clouds, Lagrangian trajectories are calculated 24 hours back in time based on hourly wind fields using LAGRANTO (Sprenger and Wernli, 2015) for each grid point containing an observed cirrus cloud. Hereby, a grid point is considered to contain a cirrus cloud if IWC > 0 and T < -38°C. Since LAGRANTO





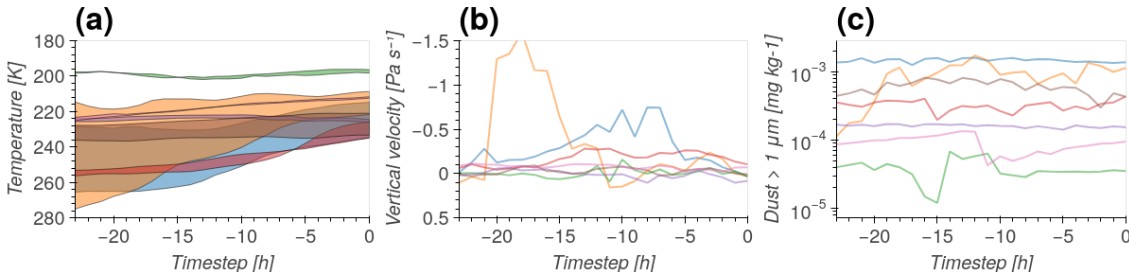

**Figure 2.** Lagrangian backward trajectories of seven randomly sampled cirrus clouds. The clouds are observed at timestep $t = 0$, from which the Lagrangian trajectories are calculated backwards in time. Each color is representing trajectories for a single cloud across the three panels. Panel (a) shows temperature along trajectories starting from cloud top and cloud base of observed cirrus, panel (b) and (c) show vertical velocity and the concentration of dust particles with radii $> 1$ µm along the trajectory at the cloud center, respectively. Note that for clouds extending to T $> -38$°C, the cloud base is defined as the last layer with T $< -38$°C at $t = 0$.

calculates trajectories based on ERA5 wind fields (Hersbach et al., 2018), DARDAR cirrus observations are first regridded onto the ERA5 grid of 0.25°×0.25° and a vertical resolution of 300 m, which approximately corresponds to the model layer thickness in ERA5 at cirrus altitudes. After regridding, only grid points with a fractional cloud cover $> 0.1$ are considered to be cirrus cloud observations. Blue dots in Fig.1 mark the trajectories for 10.000 randomly sampled cirrus (red dots). We trace temperature and large-scale vertical velocity from ERA5 along the trajectories. Cloud properties available from ERA5 are

intentionally omitted in this work, since IWC is substantially underestimated compared to DARDAR, especially in the cirrus regime (Duncan and Eriksson, 2018). The data source for dust concentration is the MERRA2 (Buchard et al., 2017; Gelaro et al., 2017) reanalysis product, which contains data for dust aerosols in five size bins. For this study we aggregate the four size bins with radii $> 1$ µm into a single super-micron size bin. Given that larger particles are more likely to act as INPs (Kanji et al., 2017), we use the super-micron dust bin as proxy for dust particle concentration for this study. In order to trace MERRA2 data

along the ERA5 trajectory, the data is regridded to the spatio-temporal resolution of ERA5 data. MERRA2 was chosen as a data source for aerosol concentration in this study as it is has been shown to better represent aerosol properties than the CAMS reanalysis product (Benedetti et al., 2009; Inness et al., 2019) when compared to remote sensing observations (Gueymard and Yang, 2020).

For the remainder of this study, we refer to a cirrus cloud as all consecutive vertical layers in a DARDAR observation containing

IWC $> 0$ mg m$^{-3}$ and located at T $< -38$°C, instead of considering each vertical level separately. This approach enables a more holistic view on the whole cloud. In order to also get a holistic view of the history for each cloud, while ensuring comparability between clouds, we define the meteorological history of a cloud by its 24-h temperature evolution along the trajectories starting at cloud top and cloud base. Correspondingly, we consider the vertical velocity and dust particle concentration along the trajectory starting at cloud center. Figure 2 visualizes these trajectories for seven randomly sampled cirrus clouds.

The domain of this study is defined from 140°W to 40°E and 0°N to 60°N and contains data from 2007 - 2009. The chosen domain covers a wide range of climatological regimes and surface types and can be considered representative for tropical and



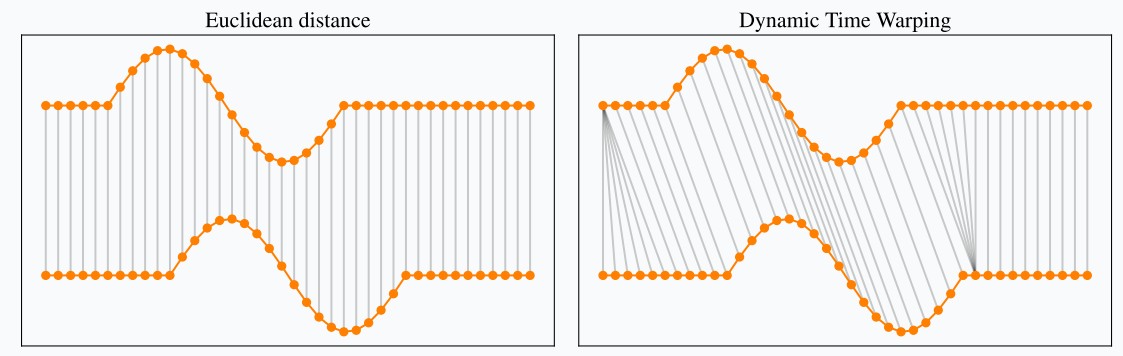

**Figure 3.** Comparison between Euclidean distance and DTW for two time series that are shifted along the temporal axis. Reprinted from Tavenard (2021)

midlatitudinal regions, where cirrus clouds occurring at latitudes $\leq$ 30°N are considered tropical and cirrus clouds occurring at latitudes > 30°N are considered midlatitudinal. The dataset contains 1.1 million cirrus cloud observations with 7.2 million separate cloud layers, for each of which a 24 h Lagrangian backward trajectory has been calculated. As cirrus clouds typically do not persist more than 12 hours (Lohmann et al., 2016), it can be assumed that the observed cirrus clouds have formed within the time frame of the 24 h trajectories. This is also in line with Jeggle et al. (2023) who found that the last 15 h of a trajectory contain all information for predicting cirrus cloud properties.

## 2.2 Trajectory Clustering

To identify different formation regimes of cirrus clouds in our dataset, we follow a data-driven approach by applying k-means clustering using dynamic time warping (DTW) as distance metric (Sakoe and Chiba, 1978). The algorithm separates time series (i.e. Lagrangian backward trajectories) into $k$ clusters where similar trajectories are grouped together in the same cluster. Instead of using point-to-point comparisons to calculate the similarity between time series using Euclidean distance, we apply DTW to account for time shifts in the backward trajectories. DTW is a similarity measure that calculates the Euclidean distance on temporally aligned time series, aiming at an improved similarity assessment compared to a point-to-point comparison. Figure 3 visualizes the differences between standard Euclidean distance and DTW. In both cases the calculated similarity is the sum of distances between matched points. It can be seen that DTW is invariant to the temporal shift of the two time series by aligning the sinusoidal pattern of the two time series along the temporal axis. We use the Python library *tslearn* (Tavenard et al., 2020) for the k-means clustering of time series with DTW and refer to its documentation for further details.

To improve the readability of this paper, the following nomenclature is defined:

- **temperature trajectory**: ERA5 temperature along a Lagrangian trajectory (as shown in Fig. 2 (a)).

- **vertical velocity trajectory**: ERA5 vertical velocity along a Lagrangian trajectory (as shown in Fig. 2 (b)).



- **dust trajectory**: MERRA2 dust concentration along a Lagrangian trajectory (as shown in Fig. 2 (c)).

- **trajectory start:** Starting point of a Lagrangian backward trajectory, i.e. $t = 0$. At this point a DARDAR cirrus cloud
observation is available.

In a first step, temperature trajectories starting at cloud top and cloud base are clustered, resulting in $k_T$ clusters per region, which are associated with respect to their ice origin. For instance, if both cloud top and cloud base trajectories remain at temperatures below -38°C, the cluster can be considered to be formed in situ in the ice phase. To differentiate updraft dependent regimes, cirrus clouds in each temperature regime are clustered to $k_\omega$ regimes by clustering along the vertical velocity trajectory
starting at the center of the cloud. This two-step clustering approach leads to the identification of cirrus clouds which have formed in similar meteorological conditions. Given the assumption that an air parcel's history containing a cirrus cloud provides information about the cloud formation and development mechanisms, our approach allows an analysis of observed cloud snapshots with respect to their formation history, which was formerly only possible for model and reanalysis data. Cirrus formation regimes may differ depending on the region in which they occur, and we therefore apply the described clustering
approach for tropical cirrus and midlatitudinal cirrus separately. By grouping cirrus of similar temperature and vertical velocity pathways, the meteorological dependence of cirrus CMP (IWC,$N_{ice}$) can be disentangled from aerosol-cloud interactions.

### 2.3   Dust effect quantification

To estimate the sensitivity of IWC and $N_{ice}$ in the identified formation regimes, a multivariate linear regression model is fitted for both variables in each formation regime. The required input to a linear regression is a single data point rather than a time
series (i.e. dust along trajectory), which is why we take the dust particle concentration at the time of cirrus cloud observation ($t = 0$) as a proxy for the dust concentration along the trajectory. The small mean standard deviation of 0.14 log mg kg$^{-1}$ along trajectories is supporting the validity of this simplifying assumption. We are interested in analyzing possible distribution shifts of cirrus CMP caused by varying dust particle concentrations, rather than the effect of dust on individual clouds. Hence, we create dust concentration bins with a width of $\frac{1}{5}$ of an order of magnitude and calculate median IWC and $N_{ice}$ values, which
are used as target variables for the regression. The width of the dust concentration bins is chosen to account for relatively small variations in dust concentrations while guaranteeing a sufficient number of cirrus observations per bin.

While having reduced the dependence of cloud ice properties on the formation regime and meteorological conditions through the trajectory clustering, it is still necessary to adjust for meteorological dependencies within each cluster. Therefore, in addition to dust concentration as a regressor, temperature ($T$) and large-scale vertical velocity ($\omega$) at the time of the observation ($t = 0$)
are also used as regressors, resulting in the following multivariate linear regression equations:

$$\text{IWC} = \beta_1^{\text{IWC}} T + \beta_2^{\text{IWC}} \omega + \beta_3^{\text{IWC}} Dust + \epsilon \tag{1}$$

$$\text{N}_{\text{ICE}} = \beta_1^{\text{N}_{\text{ICE}}} T + \beta_2^{\text{N}_{\text{ICE}}} \omega + \beta_3^{\text{N}_{\text{ICE}}} Dust + \epsilon \tag{2}$$



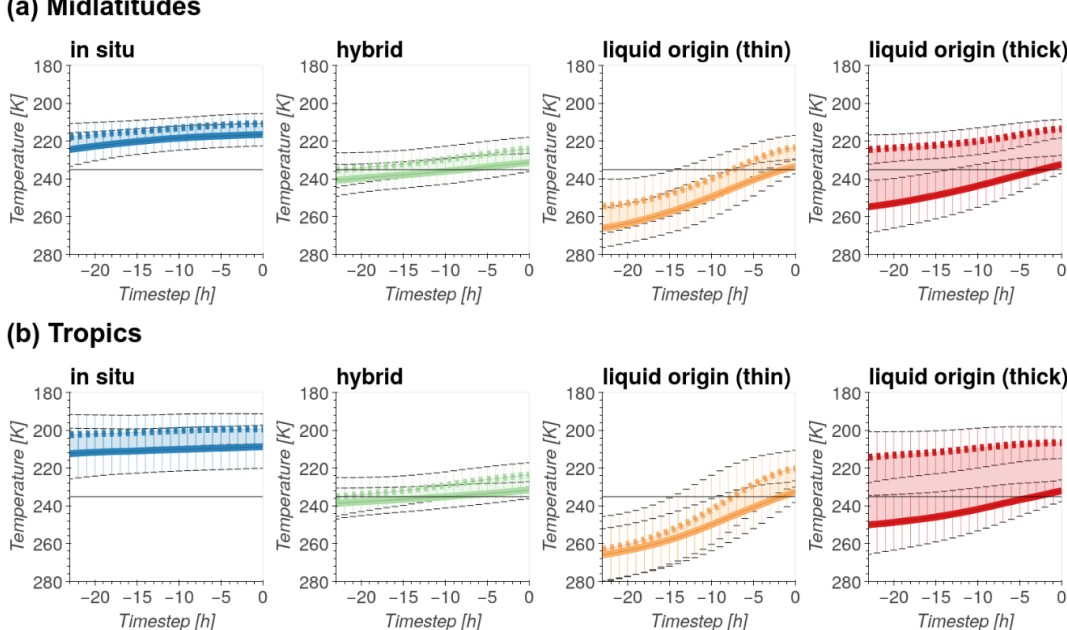

**Figure 4.** Mean temperatures along 24 h backward trajectories for four identified clusters of cirrus clouds in the midlatitudes (a) and tropics (b). t = 0 indicates the starting point of the backward trajectories, that is, the DARDAR observed cirrus clouds. Dashed lines indicate trajectories at cloud top and solid lines trajectories at cloud base. The error bars represent the standard deviations at each timestep. The horizontal black line at 235 K marks the homogeneous freezing threshold, and hence the shift from the mixed-phase to the cirrus regime.

Once fitted and the significance of the regression coefficients is tested ($p < 0.05$), the coefficients for $Dust$ can be interpreted as the sensitivity of IWC and $N_{ice}$ to a unit change in the concentration of dust. Note that throughout this study, a unit change in dust concentration corresponds to a change of one order of magnitude, as we use log-transformed concentrations.

## 3 Data-driven formation regime identification

### 3.1 Temperature regimes

By clustering the temperature trajectories as described in section 2.2, four distinct trajectory pathways can be identified. We refer to appendix A for a discussion on the choice of $k$ clusters. Each cirrus cloud in the dataset is assigned to the cluster with the minimum distance of its temperature trajectory at cloud top and cloud base. Figure 4 shows the mean temperature trajectories of cirrus clouds belonging to the four clusters for cirrus clouds in the midlatitudes (Fig. 4a) and tropics (Fig. 4b). Given the characteristics of the trajectories, the clusters are named according to their ice origin. The trajectories of the *in situ* cluster (blue color) stay at temperatures below 235 K along the whole trajectory. Assuming that the observed cirrus at t = 0 formed at any point in the 24 h prior to its observation, we are confident that cirrus belonging to this cluster formed directly in





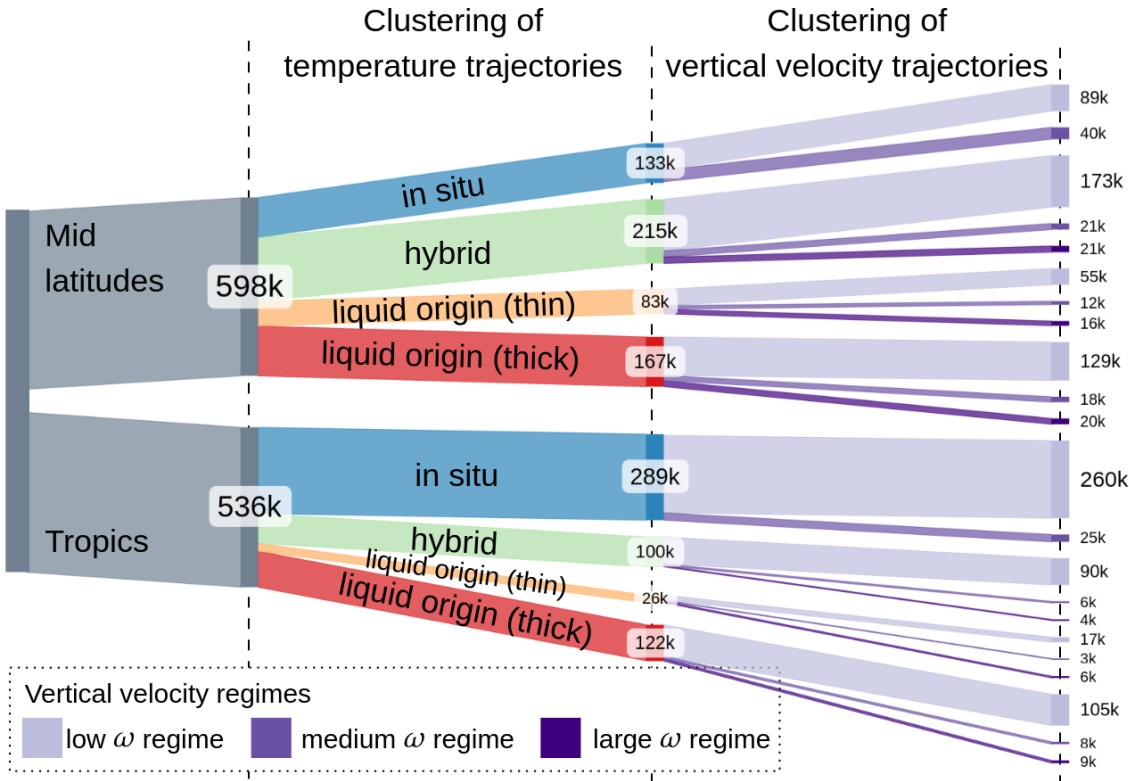

**Figure 5.** Sankey plot visualizing the classification of 1.134.000 (1134k) cirrus cloud observations into formation regimes based on k-means clustering of Lagrangian trajectories. The number of clouds contained in each cluster is proportional to the thickness of each link in the Sankey diagram and is also displayed at each node. In a first step cirrus clouds are divided by region, next reanalysis trajectories are clustered by temperature pathways, and finally each temperature cluster is further clustered into vertical velocity ($\omega$) regimes. Temperature trajectories are named with respect to the ice origin of the corresponding cirrus clouds. $\omega$ regimes are named according to the magnitude of $\omega$ along the trajectories. Note that the $\omega$ clusters are not comparable between the temperature clusters.

the ice phase, i.e. in situ. Due to the increased height of the tropopause in the tropics, in situ origin cirrus can occur at higher altitudes, resulting in a shift to colder temperatures of the mean cloud top temperatures of approximately 15 K compared to the midlatitudes. The second cluster (green color) may contain both in situ and liquid origin cirrus, given that its trajectory is gradually surpassing the homogeneous freezing temperature threshold of 238 K. Whether clouds belonging to this cluster have formed in situ or have ascended from the mixed-phase depends on the time of cloud formation. Since the moment of cloud formation is unknown, and hence the ice origin is ambiguous, this cluster is named *hybrid*. The temperature trajectories and cloud vertical extent of the remaining two clusters suggest that cirrus clouds belonging to these formation regimes are likely to have formed in the mixed-phase regime and are hence of liquid origin. Due to the difference in vertical extent, the clusters are named *liquid origin (thin)* (orange color) and *liquid origin (thick)* (red color). Figure 5 visualizes the absolute occurrence of cirrus clouds in the dataset that belong to each cluster for both midlatitudes and tropics.



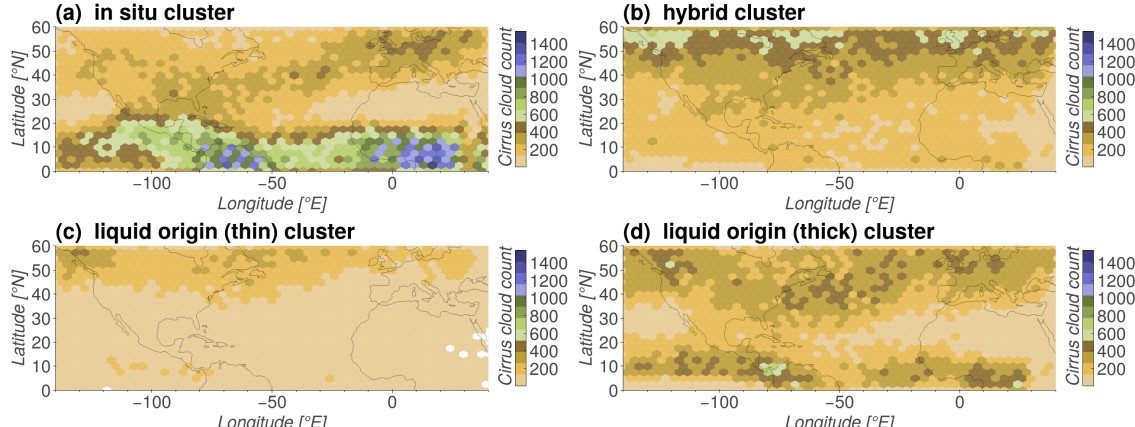

**Figure 6.** Occurrence of observed cirrus clouds for each temperature trajectory cluster on a 2.5°x2.5° grid for the years 2007 - 2009.

Fig. 6 visualizes the spatial distribution of cirrus clouds belonging to each cluster in the study domain. Cirrus clouds of the *in situ* and *liquid origin (thick)* cluster occur most often in the intertropical convergence zone (ITCZ) and in the northern Atlantic and Europe. A majority of cirrus belonging to the *hybrid* cluster occur at latitudes between 40°N - 60°N. The *liquid origin (thin)* cirrus clouds primarily occur at latitudes > 40°N and are the least frequently observed cirrus with an overall occurrence < 10%.

Cirrus cloud properties are typically characterized as a function of temperature, hence we analyze and compare the temperature dependence of the observed cloud ice properties for the identified temperature trajectory clusters. Figure 7 shows the median values of IWC (Fig. 7 a,c) and $N_{ice}$ (Fig. 7 b,d) for 1K temperature bins for each cluster as well as the median values for all clouds (dashed black line). Consistent with Krämer et al. (2016), we observe higher IWC for liquid origin cirrus compared to in situ cirrus with an increasing spread at higher temperatures, which means that the effect of ice origin on cirrus

CMP decreases with temperature and therefore altitude. At 220 K, for instance, *liquid origin (thick)* cirrus have a median IWC that is twice as large compared to median *in situ* cirrus, in both the tropics and midlatitudes. Notably, median IWC values are decreasing for *in situ* cirrus at temperatures > 215K in the midlatitudes. We assume that updrafts for cirrus in this temperature regime are too small to nucleate new ice crystals, leading to the reversal of the normally positive IWC - temperature dependence. Low median $N_{ice}$ values < 0.05 cm$^{-3}$ (Fig. 7 b) support this hypothesis.

The same effect can be observed in a weaker form in the tropics. Generally, a stronger positive IWC - temperature dependence can be observed for cirrus of liquid origin, while in situ cirrus exhibit a stronger negative $N_{ice}$ - temperature dependence. The relative temperature dependence is exemplarily quantified in Table 1 as the relative change in median IWC and $N_{ice}$ values for cirrus occurring at 210 K compared to cirrus occurring at 220 K. $N_{ice}$ decreases by 48 % (56 %) between 210 K and 220 K for in situ cirrus in the midlatitudes (tropics) probably due to a scarcity of INPs at higher temperatures. Liquid origin cirrus clouds

have a six times lower dependency of $N_{ice}$ on temperature compared to in situ cirrus clouds. As liquid origin cirrus clouds form by glaciating cloud droplets of mixed-phase clouds that are lifted to the cirrus temperature regime, it cannot necessarily





**Table 1.** Relative change of IWC and $N_{ice}$ median values from 210 K to 220 K for in situ and liquid origin (thick) clusters for cirrus in tropics and midlatitudes

|  | Midlatitudes | | Tropics | |
|---|---|---|---|---|
|  | in situ | liquid origin (thick) | in situ | liquid origin (thick) |
| IWC | - 13 % | 41 % | 22 % | 45 % |
| $N_{ice}$ | - 48 % | - 8 % | - 56 % | - 9 % |

be expected to observe larger ice crystal concentrations with decreasing temperatures. The fact that, although only slightly, $N_{ice}$ is increasing with decreasing temperatures for liquid origin clouds suggests the nucleation of new ice crystals through INPs enabled heterogenous freezing or homogeneous freezing of solution droplets triggered by cloud top cooling (Hartmann

et al., 2018). Another possible explanation for the negative temperature dependence of $N_{ice}$ for liquid origin cirrus clouds is that clouds observed at warmer temperature are aged clouds (e.g. aged anvil cirrus) that have sedimented to lower (i.e. warmer) altitudes (Doswell, 1985). Since we do not have information about the time of cloud formation, it is impossible to disentangle the described effects.

Clouds belonging to the *hybrid* cluster sit in the middle between *liquid origin (thick)* and *in situ* clusters, supporting the

hypothesis that both in situ and liquid origin cirrus clouds belong to this cluster. Given the low amount of *liquid origin (thin)* clouds in the dataset, they are not considered in further analysis.

## 3.2  Vertical velocity regimes

For each temperature-based formation regime identified in section 3.1, 2 - 3 regimes are identified by clustering the trajectories

by vertical velocity starting at the center of each cirrus cloud in the dataset. Figure 8 shows the mean vertical velocities along the 24 h trajectory for each vertical velocity regime in midlatitudes. An analog figure for tropical cirrus is shown in the supplementary Fig. B1. Each row of panels in Fig. 8 represents the vertical velocity regimes for a single temperature regime. For simplicity, the vertical velocity regimes are named *low $\omega$ regime*, *medium $\omega$ regime*, and *high $\omega$ regime* for each temperature cluster. We note that the vertical velocity regimes and the resulting cloud properties of different temperature regimes are not

directly comparable given the difference in absolute vertical velocities. For instance, the *low $\omega$ regime* of the liquid origin clusters has significantly higher updraft velocities than the *low $\omega$ regime* of the in situ cluster.

Fig. 5 visualizes the amount of cirrus observations belonging to each $\omega$ regime. The large majority of observed cirrus clouds belong to the *low $\omega$ regimes* with comparably low and steady vertical velocities along the 24 h trajectories. For in situ cirrus another vertical velocity regime was identified with slightly higher, but still steady $\omega$ values along the trajectory. Keeping the

temperature the same, the increase in vertical velocity results in higher median values of $N_{ice}$ IWC (Fig. 8 c, d). For the *hybrid*, *liquid origin (thin)*, and *liquid origin (thick)* temperature clusters a vertical velocity cluster with similar shape is identified. The trajectory experiences high vertical velocities at the beginning of the trajectory which are decreasing towards the time



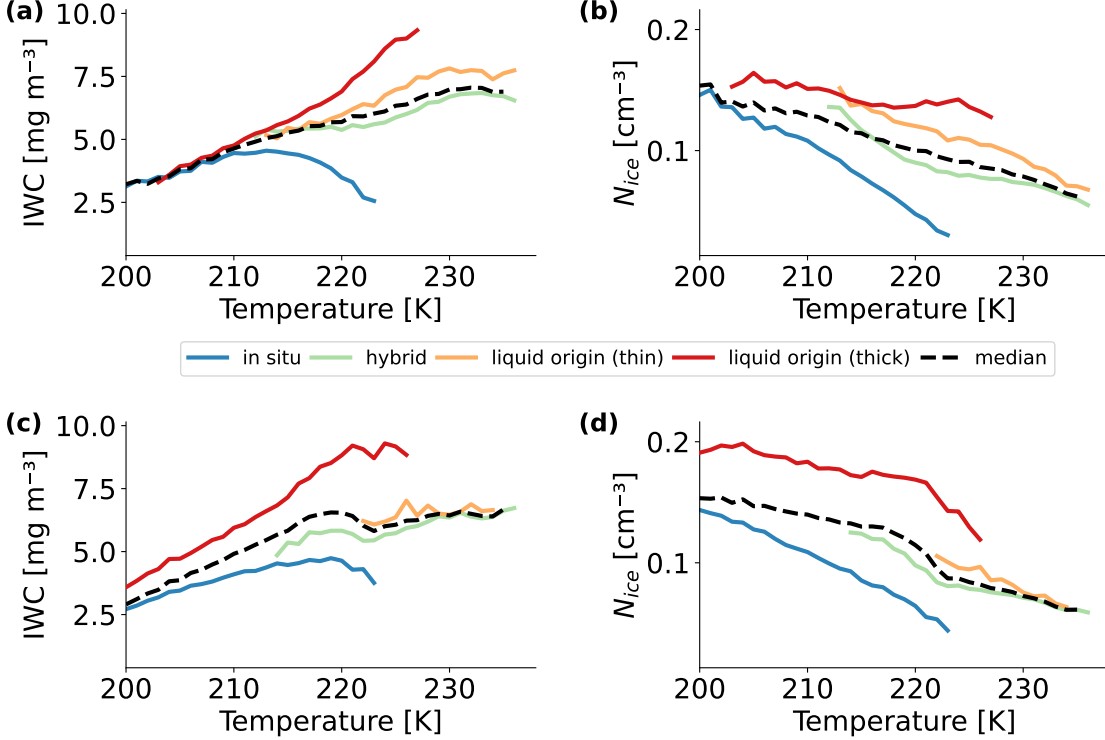

**Figure 7.** Median IWC (a) and $N_{ice}$ (b) in cloud layers with a distance to cloud top $< 600$ m of temperature trajectory clusters calculated for 1K temperature bins in the midlatitudes. (c) and (d) are analogous for the tropics. The dashed black lines represent the median value for cirrus clouds in a given region. Median values are shown for 1K temperature bins containing $> 2500$ cirrus cloud observations. The temperatures represent the ERA5 temperature at the trajectory start points ($t = 0$).

of cirrus observation (Fig. 8 f, k, p). The temporal evolution of vertical velocities indicates mature liquid origin cirrus which have formed under conditions of high vertical velocities, probably 10 - 20 hours prior to the observed cirrus. However, it is

impossible to exactly date the actual time of cloud formation due to the limited temporal resolution of the satellite data. For both *hybrid* and *liquid origin (thick)* cirrus median IWC and $N_{ice}$ values of the *medium $\omega$* regimes are very similar to the *low $\omega$* regimes, indicating that the effect of high vertical velocities during cloud formation has dissipated at the time of observation of the cirrus cloud. In the *high $\omega$* regimes (Fig. 8 g, l, q) the vertical velocities peak approximately ten hours before the observation with increased vertical velocities until the time of the observation ($t = 0$). For most temperature bins, median IWC (Fig. 8 h,

m, r) and $N_{ice}$ (Fig. 8 i, n, s) are larger in the *high $\omega$* regime compared to the other two regimes. Considering convective cloud regimes, the cirrus clouds associated with *medium $\omega$* regimes are likely aged anvil clouds and cirrus clouds associated with the *high $\omega$* regimes rather freshly detrained anvils closer to the convective core.



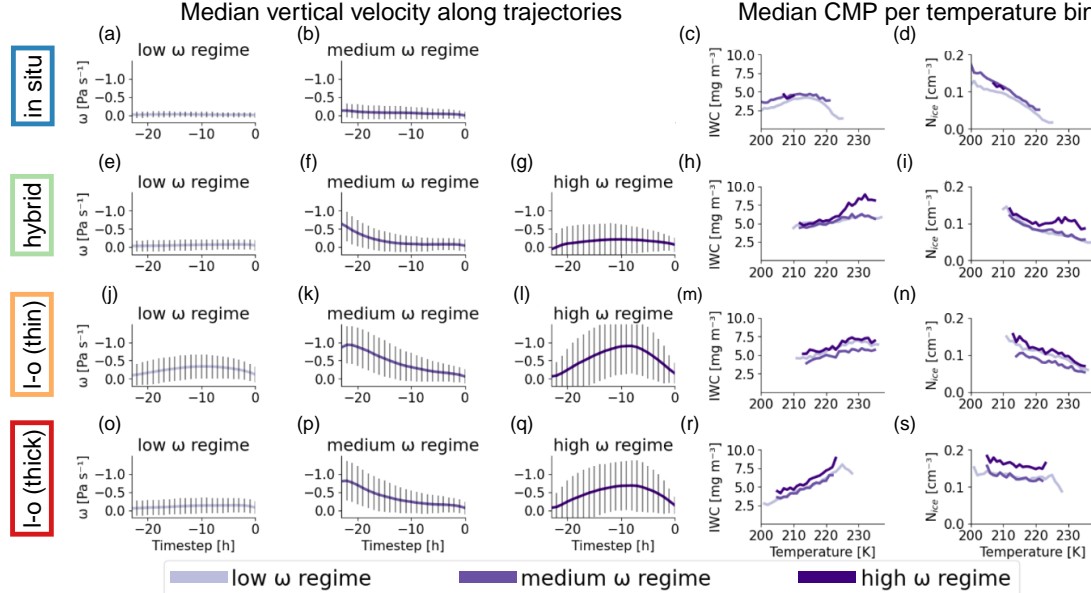

**Figure 8.** Vertical velocity regimes in midlatitudes obtained by clustering vertical velocity along Lagrangian trajectories of observed cirrus clouds. The clustering is conducted for each temperature cluster separately, resulting in three distinct vertical velocity clusters for each temperature cluster with the exception of in situ cirrus, for which two vertical velocity clusters are identified. Panels in the first three columns visualize the mean vertical velocity trajectory at the cloud center for a single regime. The error bars indicate $1\sigma$ deviation from the mean. The panels in the two rightmost columns represent the median values of IWC and $N_{ice}$ as functions of the temperature for each vertical velocity regime. Median values are shown for 1K temperature bins containing $> 500$ cirrus cloud observations. Liquid origin clusters are abbreviated by *l-o* in the figure.

## 4 Quantification of dust aerosol effect

Our goal is to isolate the effect of dust on cirrus cloud properties from other dependencies such as temperature and formation
regime. By classifying cirrus into different trajectory clusters, we can analyze the effect of dust on cirrus clouds for different cloud regimes and at the same time reduce the effect of ice origin on IWC and $N_{ice}$. Figure 9 d and h show the distributions of dust particle concentration for the different temperature trajectory clusters for midlatitudes and tropics, respectively. It can be seen that the dust concentration distributions span over four orders of magnitude. Fig. 9 a, b, e, f show median IWC and $N_{ice}$ as a function of dust particle concentration. Following the approach described in section 2.3, we analyse if the amount of available
dust has an influence on observed cirrus CMP. While clouds in the same regime occur at similar meteorological conditions, there is still a spread in both temperature and vertical velocity. As dust concentrations generally decrease with height, and therefore temperature (Fig 9 c, g), and cloud ice properties are functions of temperature (Fig. 7), the remaining meteorological effect must be taken into account when considering the influence of dust on the CMP of cirrus.





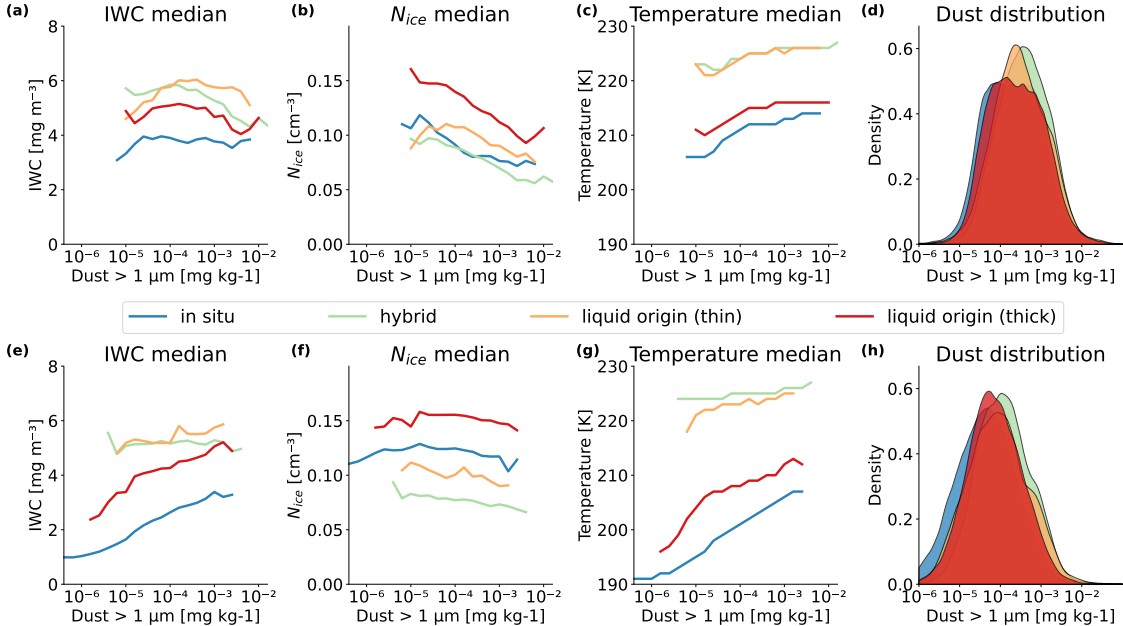

**Figure 9.** Median IWC (a), $N_{ice}$ (b), and temperature values (c) for given dust bins in the midlatitudes. Each dust concentration bin represents $\frac{1}{5}$ of an order of magnitude. Median values are shown for data points with $> 1000$ cirrus cloud observations per dust bin. Panel (d) shows the density plot of dust particle concentration for different cirrus clusters in the midlatitudes. Panels e,f,g,h are the same plots for tropical cirrus clouds.

To account for the remaining dependency of cirrus properties on temperature and vertical velocity, we conduct a multivariate
regression onto median IWC and $N_{ice}$ values with temperature, vertical velocity, and dust concentration as regressors (see eq. 1 and eq. 2). Since ice nucleation occurs at cloud top, we expect the largest effect of dust acting as INPs close to cloud top and hence use data from the cloud top layer in the regression. Fig. 10 a and b show the regression coefficient for dust ($\beta 3$ in eq. 2) in the regression onto IWC and $N_{ice}$, respectively. The regression coefficients can be interpreted as the sensitivity of the median values of IWC and $N_{ice}$ to a unit change in dust particle concentration. Grey fields in Fig. 10 indicate that the dust concentration
did not have a significant effect on the target variable ($p < 0.05$). Panels c and d show the sensitivity of IWC and $N_{ice}$ to dust as a percentage of the median IWC and $N_{ice}$ values.

As expected, $N_{ice}$ is more sensitive to the amount of dust aerosol than IWC, with all regression coefficients except for the in situ cirrus in the midlatitudes being significant. An increased availability of dust aerosols can lead to a regime shift from homogeneous freezing of solution droplets or cloud droplets to heterogenous freezing catalyzed by the availability of INPs in
the form of dust aerosols. Depending on the region and formation regime this regime shift leads to a decrease in $N_{ice}$ between 5% and 18%, which results in an increase in ice crystal size. The effect is strongest in the midlatitudes for cirrus clouds of the *hybrid* and *liquid origin (thick)* regime. Due to the sedimentation of larger ice crystals, the regime shift in the ice nucleation mechanism can lead to a reduction of IWC, as can be observed for cirrus clouds in *hybrid* and *liquid origin (thick)* regimes



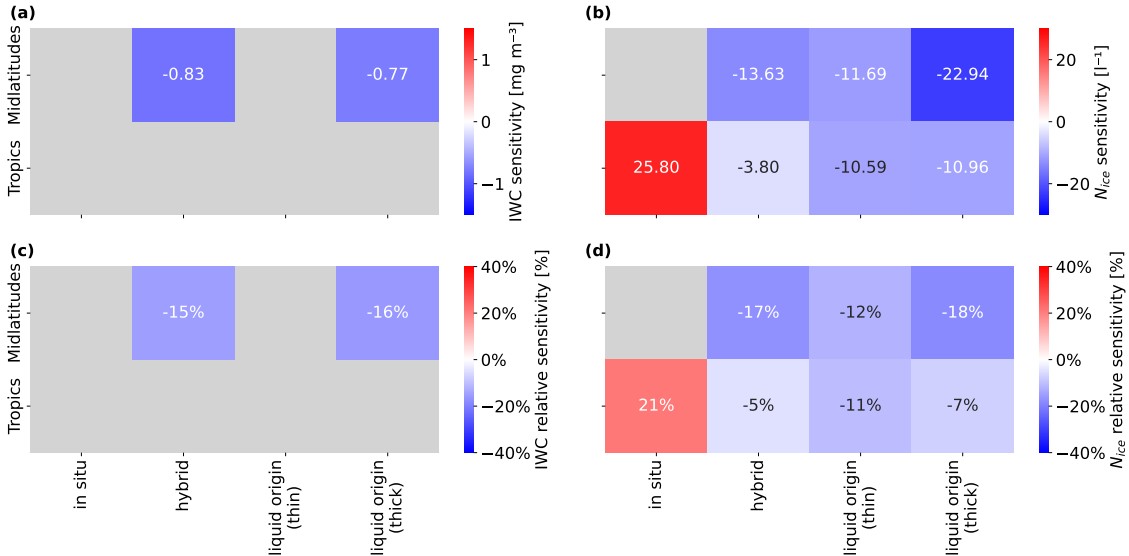

**Figure 10.** Regression coefficients $\beta_3$ in eq. 1 (a) and eq. 2 (b) representing the sensitivity of IWC and $N_{ice}$ to a unit change of dust aerosol in logarithmic space. Panels c and d represent the sensitivity as a percentage of the median IWC and $N_{ice}$ values, respectively, for each region / cirrus cluster combination. The grey fields indicate that $\beta_3$ has a p-value > 0.05, and is therefore considered statistically insignificant.

in the midlatitudes. For liquid origin cirrus the suppression of homogeneous ice nucleation by heterogeneous ice nucleation
occurs with a temporal delay. An increased availability of INPs leads to an increase of heterogeneous ice nucleation while
the cloud is still in the mixed-phase regime. This leads to a growth of ice crystals at the expense of cloud droplets via the
Wegener-Bergeron-Findeisen process (Wegener, 1911) or even full glaciation of the cloud at temperatures > -38°C and ulti-
mately resulting in less homogeneous ice nucleation and thus fewer ice crystals at cirrus temperatures (T < -38°C).

For in situ cirrus in the tropics a substantial positive sensitivity of 21% can be observed, in contrast to the negative dust
aerosol sensitivities in all other regimes. We assume that this effect is caused by heterogeneous nucleation of ice in regions
with supersaturations between the heterogeneous and homogeneous freezing thresholds and low updraft velocities, i.e. in
conditions that are INP limited and in which ice nucleation would not have happened homogeneously. Aircraft-based in situ
measurements of relative humidities in fact show that these conditions are frequently occurring at high altitudes, especially in
the tropics at temperatures below 200 K (Krämer et al., 2020).
Our findings provide observational evidence that an increased abundance of INPs in cirrus regimes, can lead to an *overseeding*
effect for CCT. Meaning, that instead of an INP induced shift from homogeneous to heterogeneous ice nucleation, more
numerous ice crystals form heterogeneously in conditions where homogeneous ice nucleation would not have occurred. Our
results confirm findings from studies with global climate models that have found an *overseeding* effect (Storelvmo et al., 2013;
Penner et al., 2015; Gasparini and Lohmann, 2016; Gasparini et al., 2017; Tully et al., 2022).



We note that with our simplified assumption of linearity between median IWC and $N_{ice}$ values and the regressors, we might miss non-linear effects and are also only able to retrieve a noisy estimate given different microphysical processes with opposing effects. For example, despite the overall positive dust sensitivity for in situ cirrus in the tropics, there may still be some in situ cirrus in the tropics that are subject to a regime shift from homogeneous to heterogeneous nucleation due to an increase of 315   dust concentration leading to a reduction of $N_{ice}$. While being able to explain the observed sensitivities with our theoretical understanding of microphysical processes, the identified sensitivities do not necessarily constitute causal effects and could also be confounded by other factors.

## 5  Limitations

Cloud-scale updrafts are a core driver of cirrus cloud formation and have a substantial influence on their CMP. Like all other 320   studies working with satellite and reanalysis data, this work is limited to the use of proxies for cloud-scale updrafts. While ERA5 can represent large-scale vertical velocity, small-scale updrafts like gravity waves and cloud-scale convection are not resolved. Additionally, uncertainties in ERA5 are higher in the tropics due to the frequent occurrence of convection, which, in turn, increases the uncertainty of our findings in the tropics. It is likely that a portion of cirrus clouds that we have classified as in situ cirrus in the tropics are in fact liquid origin cirrus formed by convection that is not resolved in ERA5. Analogue to 325   uncertainties in meteorological variables, aerosol concentrations obtained from MERRA2 are associated with region-dependent uncertainties and may not necessarily represent the actual occurrence of dust. While DARDAR IWC and $N_{ice}$ agree well with co-located in-situ observations of cirrus clouds (Krämer et al., 2016; Sourdeval et al., 2018; Krämer et al., 2020), uncertainties associated with measurement errors of the satellite instruments as well as the retrieval algorithms remain. Due to the absence of observational information about the time of cloud formation, cirrus clouds that have recently formed and cirrus clouds that 330   are in a dissolving stage may belong to the same formation regime identified in this work, resulting in noisier results. If we were able to accurately determine the time of cloud formation, a more fine-grained classification of cirrus clouds could be conducted, leading also to an increase of the signal to noise ratio in the dust effect quantification.

## 6  Conclusions

In this study, we combined three years (2007 - 2009) of vertically resolved satellite retrievals from DARDAR-Nice of cirrus 335   clouds in the domain from 140°W to 40°E and 0°N to 60°N with Lagrangian 24h backward trajectories of meteorological and aerosol variables from reanalysis data, resulting in 1.1 million cirrus clouds and their corresponding trajectories. Extending satellite data with trajectories enables an evolutionary perspective on observed cirrus clouds in contrast to the usual snapshot perspective studied with satellite data.

By clustering cirrus trajectories with k-means and DTW as distance metric, we identified four main formation regimes (*in situ*, 340   *hybrid*, *liquid origin (thin)*, *liquid origin (thick)*) with 2 - 3 vertical velocity sub-regimes each.

Consistent with existing studies, we confirm increased IWC and $N_{ice}$ values for liquid origin cirrus compared to in situ cirrus.



However, in contrast to existing research that classified cirrus cloud ice-origin in reanalysis data or in situ observations of single case studies, our method enables the classification of cirrus clouds based on their formation pathway in a large-scale satellite dataset. Furthermore, we find that IWC of liquid origin cirrus has a strong positive temperature dependence, whereas in situ
cirrus have a strong negative temperature dependence on $N_{ice}$.

The identification of cirrus formation regimes helps to disentangle the effect of dust particle concentration on cirrus cloud IWC and $N_{ice}$ from other dependencies. In line with recent evidence showing that mineral dust plays a dominant role in cirrus cloud formation (Froyd et al., 2022), we find significant sensitivities of satellite observed cirrus cloud properties, i.e. IWC and $N_{ice}$, to dust particle concentrations from MERRA2 reanalysis data. We find that increasing dust concentrations
can induce sensitivities of opposing signs caused by varying dominant microphysical processes for different cirrus cloud formation regimes. Except for tropical in situ cirrus, we detect a decrease in median $N_{ice}$ with increasing dust concentrations, with sensitivities ranging from 5% to 11% in the tropics and 12% to 18% in midlatitudinal regions per unit increase of dust concentration in logarithmic space. We attribute this decrease to a shift from homogeneous to heterogeneous ice nucleation, resulting in fewer, but larger ice crystals. Conversely, in situ cirrus in the tropics show a 21% increase in median $N_{ice}$ per
dust aerosol unit increase in logarithmic space. We assume this is caused by heterogeneous ice nucleation initiated by dust INPs in INP limited conditions with supersaturations between heterogeneous and homogeneous freezing thresholds, which are frequently found at high altitudes, especially in tropical regions. These results add observational evidence towards the the ineffectiveness and potential undesired warming effects due to *overseeding* of CCT as a climate intervention strategy.

IWC is generally less sensitive to the ice nucleation mechanism than $N_{ice}$, and hence to a change in the concentration of dust
aerosol. At midlatitudes, we find a negative sensitivity of IWC of 15 % and 16 % to dust particle concentrations for *hybrid* and *liquid origin (thick)* cirrus, likely caused by the faster sedimentation of larger ice crystals produced by heterogeneous freezing. Faster updraft velocities in the tropics counteract this effect, causing no significant sensitivities of IWC to dust particle concentrations for tropical cirrus.

To validate the effects of formation regime and dust aerosol on cirrus cloud properties, we suggest extending the study to the
southern hemisphere and additionally conducting both modeling studies and further studies exploring satellite data.

A core limitation of this work is the absence of knowledge about when an observed cloud was initially formed and where in the cloud development life cycle it was observed by the satellite. Modeling studies would enable a full evolutionary view but are limited by assumptions about cloud microphysics schemes and coarse resolutions. Studies with satellite data could include the use of geostationary data, with its high temporal resolution but with the limitation of only integrated retrievals, in contrast
to vertically resolved data.

The clustering approach applied to a dataset that combines back- trajectories of reanalysis data with vertically resolved satellite observations, introduced in this paper, may provide insights about formation regimes and aerosol-cloud interactions beyond cirrus clouds. This method could be applied to liquid and mixed-phase clouds in a similar fashion.





## Appendix A: Determining number of clusters $k$

The choice of the number of clusters $k$ is crucial for the efficacy of k-means clustering. Ideally, $k$ is chosen such that the algorithm maximizes the homogeneity of samples within a cluster and the separation of samples of different clusters. Various methods such as the "elbow" (Kodinariya and Makwana, 2013) and the silhouette method (Rousseeuw, 1987) can be used to heuristically determine $k$. The "elbow" method involves plotting the sum of squared distances of samples to their nearest cluster center as a function of the number of clusters $k$. The point where the plot bends or makes an "elbow" typically suggests

a good value for $k$, indicating a balance between compactness and separation of clusters.

   The silhouette method measures the quality of clustering by calculating the silhouette score for each sample, which is the difference between the intra-cluster distance and the nearest-cluster distance normalized by the maximum of these distances. Beyond these quantitative heuristics, domain knowledge is a key factor when determining $k$. Understanding the specific context and characteristics of the data can guide the selection of $k$, ensuring that the clusters are meaningful and relevant to the task at

hand (Jain, 2010). The computation of the silhouette scores scales quadratically with the number of samples and timesteps and is thus computationally prohibitive for our setting. To determine $k$, we combine the "elbow" method with visual interpretation and domain knowledge. Figure A1 visualizes the sum of squared distances of samples (cirrus cloud trajectories) to their nearest cluster centers as a function of $k$.

   The optimal choice of $k$ is not unambiguously identifiable according to the elbow method, but can be restricted to values between four and six. We finalize the choice of $k$ by visually interpreting the identified clusters. Figure A2 shows the identified

cluster centers for $k = 4$ (a), $k = 5$ (b), and $k = 6$ (c). It can be seen that the differentiation of additional clusters in the $k = 5$ and $k = 6$ setting does not provide much additional insight for the task of identifying cirrus formation regimes. For instance cluster 1 and 2 in Fig. A2 can be both considered in situ cirrus with similar formation conditions. Thus, $k = 4$ is chosen for the analysis presented in this study.

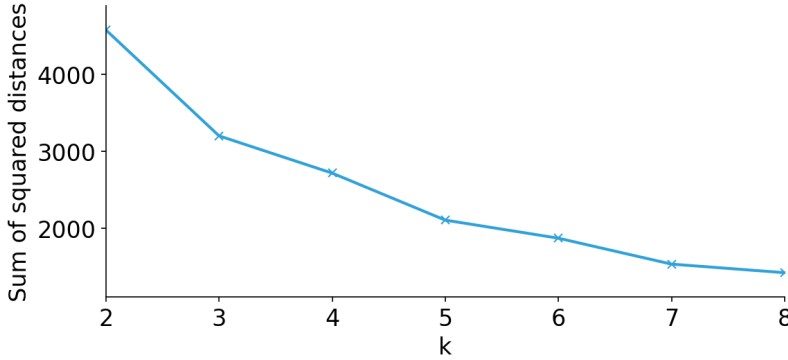

**Figure A1.** Sum of squared distances of samples (cirrus trajectories) to their nearest cluster center as a function of the number of clusters $k$.





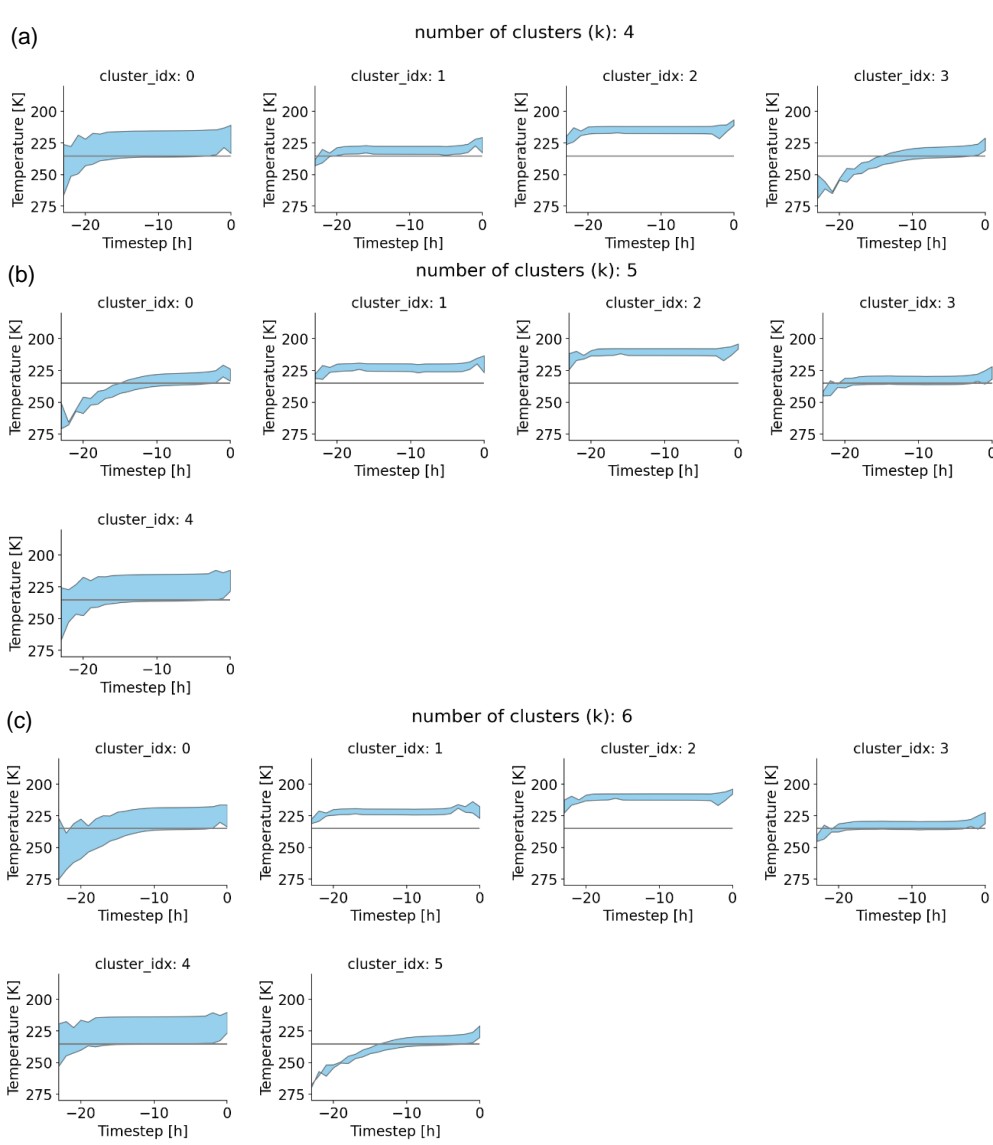

**Figure A2.** Cluster centers computed by k-means for 1.1 million cirrus cloud trajectories for $k = 4$ (a), $k = 5$ (b), $k = 6$ (c) clusters.





## Appendix B: Vertical velocity regimes - Tropics


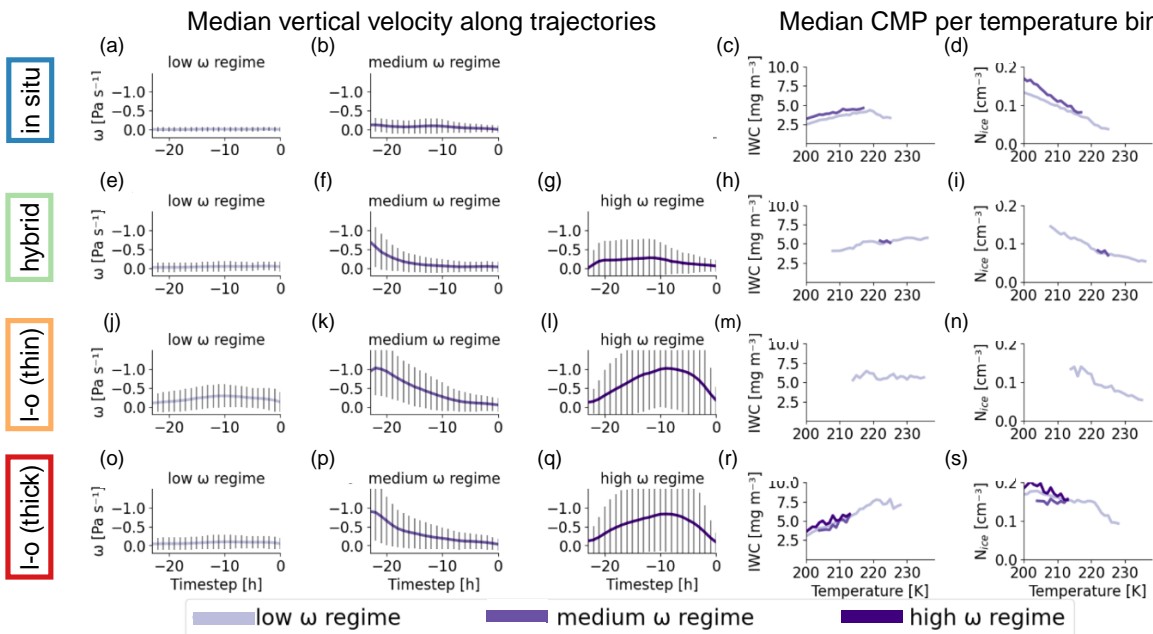

**Figure B1.** Vertical velocity regimes in the tropics obtained by clustering vertical velocity along Lagrangian trajectories of observed cirrus clouds. The clustering is conducted for each temperature cluster separately, resulting in three distinct vertical velocity clusters for each temperature cluster with the exception of in situ cirrus, for which two vertical velocity clusters are identified. Panels in the first three columns visualize the mean vertical velocity of the trajectory at the cloud center for a single regime. The error bars indicate $1\sigma$ deviation from the mean. The panels in the two rightmost columns represent the median values of IWC and $N_{ice}$ as functions of the temperature at $t = 0$ for each vertical velocity regime. Median values are shown for 1K temperature bins containing $> 500$ cirrus cloud observations. Liquid origin clusters are abbreviated by *l-o* in the figure.



*Code and data availability.* The co-located datasets, containing cirrus cloud and trajectory data, used for the analysis conducted in this paper can be downloaded from 10.5281/zenodo.13168762. The raw data can be provided by the corresponding authors upon request. The code containing the time series clustering routine and data analysis is hosted on https://github.com/tabularaza27/cloud_clustering.

*Author contributions.* KJ had the initial research idea, created the data set, developed the method and conducted the data analysis; DN and
UL supported and guided the data analysis and method development; HB provided expertise and training to calculate the trajectory data ; KJ wrote the manuscript draft; UL, DN, and HB reviewed and edited the manuscript

*Competing interests.* The authors declare that they have no conflict of interest.

*Acknowledgements.* We are grateful for the funding from the European Union's Horizon 2020 research and innovation program iMIRACLI under Marie Skłodowska-Curie grant agreement No 860100. The authors thank Heini Wernli, Michael Sprenger, and Martina Krämer for
insightful discussions, Tom Beucler for his comments on the manuscript and Odran Sourdeval for the DARDAR-Nice data.



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
