# Peer review of "Cirrus formation regimes - Data driven identification and quantification of mineral dust effect"

_EGUsphere, 2024_

## Author Comment (AC1)

**Response to EGUSPHERE-2024-2559 reviews for RC1**

We thank Reviewer #1 for their effort and feedback on our manuscript EGUSPHERE-2024-2559. In response to the suggestions and questions, please find our answers listed below: **Reviewer #1 comments are extracted in bold from original review supplement**; our responses are given directly below in normal font.

**The cirrus clouds in the liquid origin regime is formed in lower altitudes, so the aerosol concentration in lower altitudes are also important in determining N_ice in this regime. Therefore, the multiple regression equation might be revised to something as following:**

**N_ice=beta1a*T(t=0)+beta1b*T(t=-6)+ beta1c*T(t=-12)+ beta2a*w(t=0)+beta2b*w(t=-6)+ beta2c*w(t=-12)+ beta3a*dust(t=0)+beta3b*dust(t=-6)+ beta3c*dust(t=-12)+epsilon,**

**where t=0 denotes the values at DARDAR observation, t=-6 denotes values 6 hours before, and t=-12 denotes values 12 hours ago (Fig. 2).**

This is a valid point, and we considered this carefully during the manuscript preparation. The challenge lies in striking a balance between accurately modeling the dependencies of cloud properties (N_ice / IWC) and maintaining the interpretability of the model.

The small mean standard deviation of dust concentrations along the trajectory, as described in L74ff, supports the assumption that using the dust concentration at t=0 is a reasonable proxy for dust concentrations throughout the trajectory.

[L74ff: *The required input for a linear regression is a single data point, not a time series (e.g., dust along the trajectory). This is why we use the dust particle concentration at the time of cirrus cloud observation (t = 0) as a proxy for the dust concentration along the entire trajectory. The small mean standard deviation of 0.14 log mg kg−1 along the trajectories supports the validity of this simplifying assumption.*]

Reviewer #1's suggestion of extending the multiple linear regression would indeed provide more input data but would introduce challenges regarding model interpretability due to the strong autocorrelation present in time series data. To account for time series inputs, a more complex model, such as a recurrent neural network (as used by Jeggle et al., 2023), would be necessary. While such models offer improved prediction skill, they come at the cost of reduced interpretability.

Given that the focus of this work is the interpretability of the regression model, we chose a simple linear regression with a small number of input variables. This allows for regression coefficients to be directly linked to physical quantities. Furthermore, by clustering clouds into formation regimes before fitting the regression model, we can effectively disentangle the dust effect from meteorological and regional dependencies.

**The paper uses dust with size greater than 1 micron. However, although dusts with size smaller than 1 micron are less effective in acting as INP, it is possible that they**

**can still play a role. So additional analysis should be performed to demonstrate that these small dust particles are not important for N_ice.**

Thank you for this insightful point, which we also considered during the preparation of the manuscript. The decision to exclude dust particles smaller than 1 micron was based on two main reasons. First, as Reviewer #1 noted and as stated in L123, larger dust particles are more efficient ice-nucleating particles (INPs). Second, including submicron dust particles in the linear regression would likely introduce similar issues to those discussed in our previous response, specifically due to the strong autocorrelation in dust concentrations across different size ranges. Therefore, we focused on dust particles larger than 1 micron, which are more relevant to the ice nucleation process in this context.

*[L123: . For this study we aggregate the four size bins with radii > 1 µm into a single super-micron size bin. Given that larger particles are more likely to act as INPs (Kanji et al., 2017), we use the super-micron dust bin as proxy for dust particle concentration for this study.]*

**References**

Jeggle, K., D. Neubauer, G. Camps-Valls, and U. Lohmann, 2023: Understanding cirrus clouds using explainable machine learning. Environmental Data Science, 2, e19, https://doi.org/10.1017/eds.2023.14.

Kanji, Z. A., Ladino, L. A., Wex, H., Boose, Y., Burkert-Kohn, M., Cziczo, D. J., and Krämer, M.: Overview of Ice Nucleating Particles, Meteorological Monographs, 58, 1.1–1.33, https://doi.org/10.1175/AMSMONOGRAPHS-D-16-0006.1, 2017.

---

## Author Comment (AC2)

**Response to EGUSPHERE-2024-2559 reviews for RC2**

**We thank Blaž Gasparini for his effort and feedback on our manuscript EGUSPHERE-2024-2559. In response to the suggestions and questions, please find our answers listed below:** Blaž Gasparini's comments are extracted in normal font from original review supplement; our responses are given directly below in blue font; *changes in the text are given in italic plum font.*

**Overview**

The reviewers' comments and a reassessment of the clustering code led to three major changes in the manuscript:

1. **Trajectory Length Adjustment:** The cirrus cloud trajectory length was reduced from 24 hours to 12 hours to better reflect typical cirrus cloud lifetimes. Since some clouds persist beyond 12 hours, an analysis using 24-hour trajectories is included in the appendix. The results of the 12-hour trajectory clustering differ slightly from the original submission, as discussed in point 3.

2. **Removal of Vertical Velocity Clustering:** The nested clustering based on vertical velocity clusters (formerly Section 3.2) was removed to enhance the focus on the two main contributions of the paper: (i) the data-driven identification of cirrus cloud formation regimes based on temperature trajectories and (ii) the sensitivity of different clusters to dust aerosol exposure.

3. **Correction in Clustering Approach:** While rerunning the clustering for 12-hour trajectories, a bug in the data preprocessing was identified—only midlatitude cirrus clouds were used for clustering in the initial submission, and the resulting model was applied to both midlatitude and tropical cirrus. Initially, we described fitting separate clustering models for cirrus clouds in the tropics and midlatitudes (l. 170). However, after reconsidering this approach, we opted for a single clustering model trained on all cirrus cloud trajectories to improve comparability between the two regions. To still account for regional differences, the analysis of cloud properties and dust sensitivities is conducted separately for the tropics and midlatitudes.

Although these changes do not alter the core findings and conclusions, some minor adjustments in the results include:

- Instead of two liquid-origin clusters, the revised clustering identifies two temperature-separated in situ clusters.

- The sign of cloud-type sensitivity to dust aerosol exposure remains largely unchanged, though the magnitude of sensitivities has been slightly adjusted.

**General comments:**

1. The previous estimate cited in the manuscript suggests a cirrus cloud lifetime of 15 hours or less. Why are the trajectories in this study run for longer than that? Would the results change if, for example, the trajectory analysis only considered the first 10 hours?

This is a very valid remark. In response, we have reduced the trajectory duration to 12 hours. In the revised manuscript, we discuss clustering and dust aerosol sensitivities based on the 12-hour trajectories and provide a comparison to 24-hour trajectories in the appendix. The overall results remain largely unchanged, but the 12-hour analysis offers improved clarity.

2. The trajectories follow (resolved) air masses. However, formed ice crystals will also sediment out of such trajectories. Long trajectories are thus probably not very useful when trying to explain the properties of hydrometeors with considerable sedimentation rates. For instance, if my back of the envelope calculation is correct, a 40 micron (radius) ice crystal will sediment about 3 km in 10 hours. Could you comment on whether the lack of ice crystal sedimentation biases your results.

This is an interesting question. Since we have no information about the age of the cirrus clouds, we did not include estimates of ice crystal sedimentation into the methodology. We consider sedimentation in the interpretation of our results though. Our analysis is focused on cloud-top data since the coldest temperatures in a cirrus cloud and therefore ice nucleation occur mainly at cloud-top. Our results are therefore impacted by gravitational size sorting. Jensen et al. (2018) estimated the times for ice crystals <50 μm to fall 1 km in tens of hours, while ice crystals >200 μm are estimated to fall out of anvil clouds and sublimate within about 2 hours. Our data is an average over cloud-top conditions during all stages during cirrus lifetime. We expect that larger ice crystals quickly sediment from the cloud-top layer to lower cloud-layers, with the largest ice crystals falling out of the cloud and sublimating. IWC at cloud-top will be more affected than Nice, since the more numerous smaller ice crystals remain longer at cloud-top. Our results provide a conservative estimate of the effect of dust on IWC in cirrus clouds, since this gravitational size sorting provides a control over IWC at cloud-top. However, also Nice may decrease and even the whole cloud may sediment to lower altitudes (warmer temperatures) with increasing cloud age. We mention this in section 3 but since we have no information about the age of individual clouds, we cannot analyze this effect. As mentioned in section 4, the sedimentation of larger ice crystals can reduce IWC at cloud-top, which, as our results indicate, is due to a shift from homogeneous to heterogeneous ice nucleation at higher dust concentrations at cloud-top. It should also be considered that cirrus lifetimes are

much longer than the lifetimes of individual ice crystals in cirrus clouds, because cirrus ice crystals are replenished during the lifetime of cirrus clouds (see e.g., Luo & Rossow, 2004). The mechanisms responsible for the long lifetime of some cirrus clouds are still under discussion (Gasparini et al., 2023). We briefly touch upon this in section 5:

*"Knowledge about cloud age would allow also to estimate the effect of gravitational size sorting by ice crystal sedimentation. Since we focus on cloud-top data, large ice crystals may have already sedimented from cloud-top reducing IWC compared to younger clouds. $N_{ice}$ should be less affected by this unquantifiable effect of gravitational size sorting as $N_{ice}$ is dominated by the more numerous small ice crystals, which remain at cloud-top for several hours (Jensen et al., 2018)."*

3. Dust burdens come from one reanalysis, winds and other properties from another one. This seems to be another important limitation of this study, which could be avoided in new studies. Figure 2 brought this to my attention. The orange trajectory moves quickly upwards, especially at about -18 hours. At the same time, the dust concentration increases by an order of magnitude. This seems implausible.

We agree that the increase in dust along the orange trajectory shown in Fig. 2 in the original manuscript seems implausible. It likely results from using ERA5 data for meteorological variables and MERRA-2 data for dust concentrations. We now point this out in the manuscript. For the 12-hour trajectory data, there is only a slight increase in the red trajectory but the principal problem remains. However, we think our approach is reasonable for several reasons:

1) We use MERRA2 data for dust concentrations as the agreement with AERONET aerosol optical depth (AOD) is better for MERRA2 than for CAMS globally (Gueymard and Yang, 2000; see section 2.1 of the manuscript). We also use MERRA2 data for dust AOD of AERONET and CALIOP in West Asia (Gandham et al., 2022; which has some overlap with our analysis region).

2) For the quantification of the effect of dust on cirrus cloud properties, the dust particle concentration at the time of the cirrus cloud observation (t=0) is used, not the dust concentration along the trajectory. This is done because the linear regression in eqs. 1 and 2 requires a single data point as input and the standard deviation of dust along the trajectory is small (see also the answer to the first question of Reviewer #1).

3) Median temperatures as a function of dust concentration shown in Fig. 9c, g show that on average, warmer temperatures are associated with higher dust concentrations as could be expected from a decrease of temperature and dust concentrations with altitude.

4. How are different omega regimes defined? Is there a fixed omega threshold to distinguish them?

Also, although this has already been described, using units of Pa/s in a study of cloud properties doesn't seem the best, as I think the cooling rate and cloud properties care about m/s winds. Could you mention how much e.g. 0.1 m/s in omega units can vary between e.g. 10 km and 14 km?

On the other hand, I don't think that breaking down the results into vertical velocity trajectories adds much value to the study (or at least I haven't noticed it), and could therefore be moved to the appendix.

Thank you for this suggestion. Based on your comments and a reassessment of the scientific value added by the vertical velocity cluster, we have decided to remove Section 3.2 from the manuscript entirely. This allows us to sharpen the focus on the two main messages of the paper: the data-driven identification of cirrus cloud formation regimes based on temperature trajectories and the sensitivity of different clusters to dust aerosol exposure.

An overarching question related to comments 1-4:

If you or someone else were to repeat a similar study, what improvements would you make? Is LAGRANTO the best tool given that you are tracking dust (and ERA5 doesn't have dust)?

We use MERRA-2 dust aerosol mixing ratios because they show better agreement with observed dust aerosol optical depth and extinction compared to CAMS data. An alternative approach would be to use CAMS in combination with back trajectories from ERA5, as both datasets are based on IFS and should be more consistent.

LAGRANTO is a dataset agnostic tool. MERRA-2 meteorological data could be used with LAGRANTO, ensuring that both meteorological and aerosol data are MERRA-2-based. However, regardless of the choice of reanalysis dataset, significant uncertainties will remain due to unresolved subgrid processes, such as small-scale updrafts.

Beyond tracking cloud trajectories with Lagrangian methods in reanalysis data, emerging observational approaches offer new possibilities. For example, Jeggle et al. (2024) developed a 3D-resolved dataset of tropical cloud ice with a 15-minute temporal resolution by training a machine learning model to map Meteosat SEVIRI optical satellite imagery to vertical cloud structures and properties, using sparse DARDAR profiles for training. This dataset enables cloud tracking directly from satellite data, eliminating concerns about model uncertainties. However, the limitations of such novel datasets, including potential biases and observational constraints, must also be carefully considered.

**Specific comments:**

Line 15: "unit increase of dust in log space"

Maybe better order of magnitude increase in dust (?)

Updated accordingly.

Line 26:

I don't think you prove/indicate it'll increase the CRE, that seems to be more of a discussion point. In addition, the largest sensitivities to dust are in the tropical tropopause layer, probably associated with the thinnest cirrus. The change in top-of-the-atmosphere CRE may therefore be very small.

=> On the other hand, by thinking outside of the box, the study shows that by increasing dust and modifying ice nucleation in liquid-origin cirrus, the CRE can turn less positive (big question mark here as such cirrus occur at warmer temperatures, so their CRE may be on average dominated by the SW component).

We agree and changed this sentence and the preceding sentence to reflect that this overseeding may only potentially cause a positive CRE:

*"Our results provide an observational line of evidence that the climate intervention method of seeding cirrus clouds with potent INPs may potentially result in an undesired positive cloud radiative effect (CRE), i.e. a warming effect. Instead of producing fewer but larger ice crystals, we show that additional INPs can lead to an increase in $N_{ice}$ and IWC, an effect called overseeding."*

We also added a short discussion of the impact on CRE in the conclusion section:

*"Cold, thin cirrus have on average a net positive CRE due to their stronger longwave CRE compared to their shortwave CRE. An increase in these clouds could contribute to additional warming. However, their CRE may be relatively small because these clouds remain quite thin (Krämer et al., 2020)."*

Line 28: "increasing occurrence towards the equator"

Looking from pole to equator, I would say: increased occurrence until you reach the storm tracks, then decreased occurrence in the subtropics, and increased again toward the ITCZ and especially the warm pool area.

Changed as suggested.

*"When analyzing the zonal distribution from the poles towards the equator, the occurrence peaks in the mid-latitude storm tracks, followed by a decrease in the subtropics. The occurrence then increases again towards the Intertropical Convergence Zone (ITCZ), with a particularly pronounced increase over the warm pool region."*

Line 30: The definition of cirrus used in this study could be even more explicit.

To provide more clarity, we added the following sentence:

*"In this study, we define cirrus clouds as all clouds with a cloud-top temperature colder than -38°C. This definition includes, for example, the top part of deep convective cores, which are not classified as cirrus clouds in other studies. However, our clustering approach, based on back-trajectory reanalysis data, aims to capture all evolutionary stages of cirrus clouds. Therefore, we adopt this broad definition."*

Line 46: "Their counter part"  ==> that sounds weird, please use a different term.

We changed the sentence to:

*"In contrast, liquid-origin cirrus clouds evolve from mixed-phase clouds, where cloud droplets freeze heterogeneously via ice-nucleating particles (INP) (Kanji et al., 2017) at temperatures T>−38∘C"*

Lines 38-40: Some other studies classifying cirrus origin may deserve to be cited here, e.g. Muhlbauer et al., 2014 (10.1002/2013JD020035), Sassen and Comstock, 2001 (10.1175/1520-0469(2001)058<2103:AMCCCF>2.0.CO;2) use a dynamical regime classification of cirrus.

Thank you we added these references in the introduction:

*"Other studies like Sassen and Comstock (2001) and Muhlbauer et al. (2014) use a dynamical regime for the classification of cirrus."*

Lines 90-92:

I miss one sentence about the limitations of the models as a segue to the "we do this and that" part.

We added the following sentence:

*Due to uncertainties in parameterizing subgrid processes, modeling studies have inherent uncertainty. Studies based on observational data can provide additional evidence to support and validate the outcome of these model studies.*

Introduction/discussion:

It might be good to mention the aircraft observations by Mingui Diao's group. Their main conclusions, as far as I understand, point in the same direction, i.e. an increase in the number of ice for a larger number of coarse mode aerosols. See e.g. Maciel et al., 2023 (https://doi.org/10.5194/acp-23-1103-2023), or earlier work by Patnaude et al.

Thank you for this suggestion. We added a sentence in the conclusions about their interesting work:

*"Aircraft observations show that the impact of large aerosol particles like dust on IWC and $N_{ice}$ depends on the evolutionary phase of cirrus clouds (Patnaude and Diao, 2020; Maciel et al., 2023)."*

Section 3.1:

Assuming that clouds form mainly in the last 10 hours before the satellite overpass, large parts of the hybrid category would rather fit into the in-situ cirrus. On the other hand, the sensitivities make the hybrid category rather similar to the liquid-origin category. Why?

The 12-hour or 24-hour clusterings show a similar classification of cirrus clouds. Using shorter back-trajectories does not shift clouds from the hybrid to in-situ clusters. In the original manuscript the hybrid cluster showed similar sensitivities as the liquid-origin (thin) cluster. With the corrected cirrus clustering, there is no longer the liquid-origin (thin) cluster and the hybrid cluster shows sensitivities which are distinct from the other clusters.

Figure 6:

While this is addressed to some extent in the manuscript, I would like to point out that the chosen domain covers exactly the part of the world with the highest expected dust concentrations, and thus may not be representative of the more remote regions, especially in the southern hemisphere.

This is a fair point. We have added a sentence in section 5 to reflect this:

*"The study domain is a region with the highest annual mean atmospheric dust concentrations (Gavrouzou et al., 2021) and may therefore be less representative for the Southern Hemisphere or more remote regions."*

Line 235:

Gravity waves will be an even more common source of nucleation. Some references to consider Atlas and Bretherton, 2023 (https://doi.org/10.5194/acp-23-4009-2023), Chang and L'Ecuyer, 2020 (https://doi.org/10.5194/acp-20-12499- 2020), Kim et al., 2016 (https://doi.org/10.1002/2016GL069293).

Thank you. We added gravity waves as a source of nucleation in this sentence as suggested:

*"The fact that, although only slightly, $N_{ice}$ is increasing with decreasing temperatures for liquid origin clouds suggests the nucleation of new ice crystals through \ac{inp} enabled heterogenous freezing or homogeneous freezing of solution droplets triggered by cloud top cooling (Hartmann et al., 2018) or gravity waves (Kim et al., 2016; Chang and L'Ecuyer, 2020; Atlas and Bretherton, 2023)."*

Figure 7:

Are there any substantial differences between midlatitude and tropical cirrus, when binned into temperatures like here? It seems to be that one could merge the two regions. This is also a powerful qualitative representation that cirrus properties are, on average, simply controlled by thermodynamics/temperatures.

Thank you for this insightful comment. While there are no differences on the scale of orders of magnitude, we still observe substantially higher values in certain cirrus properties, such as the ice water content (IWC) of liquid-origin cirrus, in the tropics compared to the midlatitudes.

Moreover, the primary reason for differentiating between these regions is to analyze the influence of dust aerosols. Our results indicate that cirrus clouds in the tropics and midlatitudes exhibit slightly different sensitivities to dust, which justifies maintaining this distinction in our analysis.

Figure 9:

Why is the dust concentration so similar between e.g. in-situ and liquid-origin cirrus in the tropics. Their temperatures and thus altitudes are quite different (maybe a delta z of about 3 km on average). Because of that, I would expect a larger difference in dust concentrations.

Only cloud layers within 325 m from cloud top are used in Fig. 9 (new Fig. 8). We added this information in the caption of the new Fig. 8. Dust concentrations in in-situ (cold) cirrus are about 5 times lower than in in-situ, hybrid or liquid origin cirrus. We have added the median dust concentration per cirrus regime in Fig. 9c, g (new Fig. 8c, g). Below are average dust concentrations and temperatures in the tropics shown as a function of pressure for the JJA and DJF seasons of 2008. The dust concentrations corresponding to the median temperature difference between in-situ (cold) and in-situ or liquid origin are about 5-10 higher at in-situ or liquid origin median temperatures than at in-situ (cold) median temperatures. The difference in dust concentrations is indeed somewhat smaller between the cirrus regimes than what could be expected from average dust concentrations at

different altitudes. Cirrus clouds will form in regions where updraft velocities are higher than on average, which could lead to somewhat higher dust concentrations in cirrus clouds. This may explain the small difference in dust concentrations between different cirrus regimes.

[Figure]

Line 348:

However, the radiatively most relevant cirrus, at high ICNC, are homogeneously formed based on Froyd et al., 2022.

As shown in Froyd et al. (2022), homogeneously formed cirrus clouds have on average more than one order of magnitude higher $N_{ice}$. However, Froyd et al. (2022) define the radiatively most relevant cirrus clouds by $N_{ice} > 10\ l^{-1}$, and these are initiated by dust in 72% of all their analyzed cases in the Northern Hemisphere extra-tropics and 47% globally. To reflect that dust plays a dominant role in cirrus cloud formation in the Northern Hemisphere we changed the sentence to:

*"In line with recent evidence showing that mineral dust plays a dominant role in cirrus cloud formation in the Northern Hemisphere (Froyd et al., 2022), we find significant sensitivities of satellite observed cirrus cloud properties, i.e. IWC and $N_{ice}$, to dust particle concentrations from MERRA2 reanalysis data."*

Lines 366-367:

"the absence of knowledge about when an observed cloud was initially formed and where in the cloud development life cycle it was"

Any suggestion on how to do it better?

That is a very interesting avenue for future research. As mentioned earlier, recent advancements in synthetic observational data, such as the IceCloudNet dataset by Jeggle et al. (2024), could provide valuable insights into cloud development and the timing of cloud formation.

Line 369:

But maybe the information about cloud top is all what we need, assuming that this is the most important layer for ice nucleation?

Since ice crystal sedimentation/gravitational size sorting can affect IWC at cloud top, the information obtained from cloud-top may be incomplete. Considering the evolution of all cloud layers may give additional information about the impact of dust/ice nucleation on cirrus properties.

References:
Gandham, H., Dasari, H.P., Karumuri, A. *et al.* Three-dimensional structure and transport pathways of dust aerosols over West Asia. *npj Clim Atmos Sci* **5**, 45 (2022). https://doi.org/10.1038/s41612-022-00266-2

Gasparini, B., Sullivan, S. C., Sokol, A. B., Kärcher, B., Jensen, E., and Hartmann, D. L. (2023). Opinion: Tropical cirrus – from micro-scale processes to climate-scale impacts, Atmos. Chem. Phys., 23, 15413–15444, https://doi.org/10.5194/acp-23-15413-2023

Jeggle, K., Czerkawski, M., Serva, F., Saux, B. L., Neubauer, D., & Lohmann, U. (2024). IceCloudNet: 3D reconstruction of cloud ice from Meteosat SEVIRI. arXiv preprint arXiv:2410.04135.

Jensen, E. J., van den Heever, S. C., & Grant, L. D. (2018). The life cycles of ice crystals detrained from the tops of deep convection. Journal of Geophysical Research: Atmospheres, 123, 9624–9634. https://doi.org/10.1029/2018JD028832

Luo, Z. and Rossow, W. B. (2004). Characterizing Tropical Cirrus Life Cycle, Evolution, and Interaction with Upper-Tropospheric Water Vapor Using Lagrangian Trajectory Analysis of Satellite Observations, J. Climate, 17, 4541–4563

---

## Author Response (AR1)

**Cover letter to the editor of EGUSPHERE-2024-2559**

Dear Sergio Rodríguez,

We thank you and the reviewers for reviewing our paper and providing invaluable feedback, which improved our manuscript.

The reviewers comments and a reassessment of the clustering code led to three major changes in the manuscript:

1. **Trajectory Length Adjustment:** The cirrus cloud trajectory length was reduced from 24 hours to 12 hours to better reflect typical cirrus cloud lifetimes. Since some clouds persist beyond 12 hours, an analysis using 24-hour trajectories is included in the appendix. The results of the 12-hour trajectory clustering differ slightly from the original submission, as discussed in point 3.

2. **Removal of Vertical Velocity Clustering:** The nested clustering based on vertical velocity clusters (formerly Section 3.2) was removed to enhance the focus on the two main contributions of the paper: (i) the data-driven identification of cirrus cloud formation regimes based on temperature trajectories and (ii) the sensitivity of different clusters to dust aerosol exposure.

3. **Correction in Clustering Approach:** While rerunning the clustering for 12-hour trajectories, a bug in the data preprocessing was identified—only midlatitude cirrus clouds were used for clustering in the initial submission, and the resulting model was applied to both midlatitude and tropical cirrus. Initially, we described fitting separate clustering models for cirrus clouds in the tropics and midlatitudes (l. 170). However, after reconsidering this approach, we opted for a single clustering model trained on all cirrus cloud trajectories to improve comparability between the two regions. To still account for regional differences, the analysis of cloud properties and dust sensitivities is conducted separately for the tropics and midlatitudes.

Although these changes do not alter the core findings and conclusions, some minor adjustments in the results include:

- Instead of two liquid-origin clusters, the revised clustering identifies two temperature-separated in situ clusters.
- The sign of cloud-type sensitivity to dust aerosol exposure remains largely unchanged, though the magnitude of sensitivities has been slightly adjusted.

The described changes are also reflected in the accompanying code and data repositories.

Following the recommendation of the editorial support, we updated the color scheme of Figures 4, 5, 7, and 8 to allow readers with colour vision deficiencies to correctly interpret our findings.

Sincerely,

Ulrike Lohmann
Professor for Experimental Atmospheric Physics
Institute of Atmospheric and Climate Science, ETH Zurich, Zurich, Switzerland
ulrike.lohmann@env.ethz.ch